# Clinical and Molecular Presentation of a Patient with Paternal Uniparental Isodisomy of Chromosome 16

**DOI:** 10.3390/ijms26178521

**Published:** 2025-09-02

**Authors:** Elizaveta Panchenko, Natalia Semenova, Olga Sereda, Daria Guseva, Zhanna Markova, Nadezhda Shilova, Olga Simonova, Anton Smirnov, Dmitry Pustoshilov, Arina Khalilova, Vasilisa Udalova, Ilya Kanivets, Dmitry Zaletaev, Vladimir Strelnikov, Sergey Kutsev

**Affiliations:** 1Research Centre for Medical Genetics, 115522 Moscow, Russia; semenova@med-gen.ru (N.S.); anton.smirnov.9910@gmail.com (A.S.); vstrel@list.ru (V.S.);; 2Department of General and Medical Genetics, Pirogov Russian National Research Medical University, 117513 Moscow, Russia; 3Yaroslavl Regional Perinatal Center, 150042 Yaroslavl, Russia; 4Biotech Campus Limited Liability Company, 117437 Moscow, Russia; dpustoshilov@biotc.ru; 5Genomed Medical Center, 115419 Moscow, Russia

**Keywords:** imprinting disorders, uniparental disomy, chromosome 16, microsatellite analysis, chromosomal microarray analysis, FISH, whole-exome sequencing, MS-MLPA, nanopore sequencing

## Abstract

Uniparental disomies (UPDs) are among the causes of imprinting disorders. Specific phenotypes of most causative UPDs have been described. Here, we describe the case of a 2-year-old female patient who presented a syndromic phenotype. Chromosomal microarray analysis revealed UPD of the whole chromosome 16. Microsatellite analysis demonstrated paternal origin of the UPD and its isodisomic pattern (UPiD (16) pat). Mosaic trisomy 16 was not detected using the FISH method. Whole-exome sequencing revealed no pathogenetic genetic variants sufficient to explain the syndromic phenotype nor unmasked pathogenic recessive genetic variants on chromosome 16. Whole-genome trio DNA sequencing revealed no additional candidate pathogenic genetic variants to those detected by whole-exome sequencing, including miRNAs and lncRNAs. Imprinting disorders at 6q24.2, 7p12.2, 7q32.2, 11p15.5, 14q32.2, 15q11.2, and 20q13.32, as well as multilocus imprinting disturbances (MLIDs), were excluded by Methylation-Specific Multiplex Ligation-Dependent Probe Amplification (MS-MLPA). At the same time, we detected abnormal hypermethylation of the *ZNF597* transcription start site differentially methylated region (*ZNF597:*TSS-DMR), accompanied by hypomethylation of the neighbouring *ZNF597*:3′ DMR. Both DMRs were normally imprinted, and the DNA alterations in our patient with UPD (16) pat are opposite to those previously described for maternal uniparental disomy (UPD (16) mat). To date, several cases of UPD (16) pat have been reported. Our case report describes the syndromic phenotype of a patient with paternal uniparental disomy of chromosome 16 in contrast to the previously described patients with a normal phenotype or with abnormal phenotypes caused by acquired homozygosity of pathogenic variants at autosomal recessive genes located on this chromosome. Reporting such observations will help systematize data on the phenotypes of imprinting disorders on chromosome 16.

## 1. Introduction

Genomic imprinting is implemented via monoallelic DNA methylation, anomalies of which in imprinted regions of chromosomes, depending on parental origin, lead to different phenotypes of diseases. Syndromic disorders caused by disturbed human imprinting are associated with chromosome 6 (transient neonatal diabetes mellitus type 1 (OMIM# 601410); chromosome 7 (Silver–Russell syndrome type 2 (OMIM# 618905)); chromosome 8 (Birk–Barel syndrome (OMIM# 612292)); chromosome 11 (Beckwith–Wiedemann syndrome (OMIM# 130650) and Silver–Russell syndrome type 1 (OMIM# 180860) and type 3 (OMIM# 616489)); chromosome 14 (Temple syndrome (OMIM# 616222) and Kagami–Ogata syndrome (OMIM# 608149)); chromosome 15 (Prader–Willi syndrome (OMIM# 176270), Angelman syndrome (OMIM# 105830), Schaaf–Yang syndrome (OMIM# 615547), and central precocious puberty 2 (OMIM# 615346)); chromosome 16; chromosome 20 (pseudohypoparathyroidism type 1A (OMIM#103580), type B (OMIM# 603233), and type C (OMIM# 612462), pseudopseudohypoparathyroidism (OMIM# 612463), osseous heteroplasia, progressive (OMIM# 166350), and Mulchandani–Bhoj–Conlin syndrome (OMIM# 617352)) [1]. Uniparental disomies (UPDs) are among the causes of imprinting disorders [2]. The first clinical case of a UPD, diagnosed by an analysis of polymorphic DNA markers using Southern blotting, was reported in 1988 and described a girl with cystic fibrosis, short stature, and UPD (7) mat [3]. At present, most cases of loss of heterozygosity (LOH), which can be a consequence of UPD, are detected in chromosomal microarray analyses [4].

Specific phenotypes of most pathogenic UPDs have been described, which manifest depending on the parental origin of the methylation anomaly. The first case report of a patient with paternal uniparental disomy of chromosome 16 with a phenotype of bilateral pes calcaneus, an additional rudimentary mandibular dental arch, and normal physical and psychomotor development was published at 2000 [5]. Later, in 2021, a patient with paternal uniparental isodisomy and heterodisomy of chromosome 16 had a normal phenotype [6]. A specific syndromic phenotype has not been described for patients with LOH of some imprinted regions of the genome, in particular, for UPD (16) pat, in addition to those that can be explained by mutations of autosomal recessive genes like *GPT2* [7], *FA2H* [8,9], *ABCA3* [10], *FANCA* [11], *SPG35* [12], *PMM2* [13], *ALG1* [14], *GAN* [15], and *WWOX* [16] within the UPD.

Locus 16p13.3 is the most well-studied imprinted locus on chromosome 16. The paternally imprinted somatic differentially methylated region (sDMR), *ZNF597*:TSS-DMR, was identified in the shared promoter region of the *ZNF597* and *NAA60* genes, which regulates the expression of both genes. The neighbouring *ZNF597*:3′ DMR is a maternally imprinted germinal differentially methylated region (gDMR) functioning as an upstream regulator of the *ZNF597* transcription start site DMR (*ZNF597:*TSS-DMR). The biological functions of *ZNF597* and *NAA60* remain to be clarified [17,18]. Here, we demonstrate the case of a 2-year-old female patient with UPiD (16) pat who presented a syndromic phenotype.

## 2. Results

### 2.1. Clinical Presentation

The proband was an affected 2-year-old female born to non-consanguineous Russian parents. The pregnancy was complicated by anemia and pre-eclampsia. The prenatal period was complicated by fetal growth restriction at the 30th week of pregnancy. The girl was born by cesarean section (C-Section) at 37–38 gestational weeks (GWs). The birth weight was 1800 g (Z-score −3.7 SDS (standard deviation score)), the birth length was 45 cm (Z-score −2.2 SDS), and the Apgar score was 8/8. Body weight deficiency syndrome after birth was the reason for hospitalization to the neonatal intensive care unit for 27 days. The child was examined at the Research Centre for Medical Genetics. At 8 months of age, the patient’s weight was 6.15 kg (Z-score −2.33 SDS), and her length was 65 cm (Z-score −1.92 SDS). At 11 months of age, the patient’s weight was 6.5 kg (Z-score −2.61 SDS), and her length was 70 cm (Z-score −1.39 SDS). The phenotype included plagiocephaly, hypotelorism, arched eyebrows, snub nose, and proximal displacement of the thumbs. Instrumental investigations revealed an atrial septal defect, pulmonary valve stenosis, pelvic dystopia and rotation of the left kidney, assimilation of the atlas, spina bifida posterior C1, and hypoplasia of the axial atlas. At 2 years of age, the patient’s weight was 10.5 kg (Z-score −0.73 SDS), and her length was 84 cm (Z-score −0.74 SDS). During dynamic observation, the girl’s physical development indicators lagged behind age norms. Her phenotype included microcephaly, high anterior hairline, arched eyebrows, hypotelorism, epicanthus, almond-shaped palpebral fissures, wide nasal bridge, depressed nasal ridge, wide base of the nose with a broad tip, smoothed filter, full cheeks, downturned corners of the mouth, short chin, dysplastic ears, narrow funnel-shaped chest, cone-shaped fingers of the hands, proximal displacement of thumbs, hyperlordosis, valgus knees and feet, and rocker-bottom foot (Figure 1). Her speech and motor development were delayed. The Face2Gene v.6.2.6 service, accessed on 10 May 2024 [19], offered several diagnostic hypotheses, including Williams–Beuren syndrome (OMIM# 194050), Angelman syndrome (OMIM# 105830), and Prader–Willi syndrome (OMIM# 176270). Given the syndromic nature of the disease, the future management plan includes supervision by a pediatrician, neurologist, cardiologist, orthopedist, and geneticist.

### 2.2. Molecular Genetic Findings

#### 2.2.1. Loss of Heterozygosity on Chromosome 16

Chromosomal microarray analysis revealed a loss of heterozygosity (LOH) on chromosome 16. The molecular karyotype of the proband (according to ISCN 2016 [20]) was arr [GRCh37] 16p13.3q24.3 (89561_90163275) × 2 hmz (Figure 2).

#### 2.2.2. The Loss of Heterozygosity on Chromosome 16 Is Due to Paternal Uniparental Disomy

Microsatellite analysis revealed that the patient had inherited both copies of chromosome 16 from her father in an isodisomic manner (Figure 3, Figure 4 and Figure 5).

#### 2.2.3. Low-Level Trisomy 16 Mosaicism Was Excluded by FISH

No cases of trisomy 16 were detected using FISH with a chromosome 16 centromere-specific DNA probe among at least three hundred interphase nuclei and metaphase spreads from cells cultured from peripheral blood and skin fibroblasts (Figure 6).

#### 2.2.4. Absence of Detected Unmasked Pathogenic Recessive Genetic Variants on Chromosome 16

Whole-exome sequencing was performed to search for the causative genetic variants throughout the exome and homozygous autosomal recessive genes defects, especially on chromosome 16. Three pathogenic variants were detected on autosomes other than chromosome 16, *COG2* (chr1) gene heterozygous frameshift variant NM_007357: c.1034_1038delCCATA (*p*.Thr345fs), *P3H2* (chr3) gene heterozygous splice variant NM_018192:c.2034+1G>A, and *CNGB3* (chr8) gene heterozygous frameshift variant NM_019098:c.819_826delCAGACTCC(*p*.Arg274fs). No unmasked pathogenic recessive genetic variants on chromosome 16 were detected.

Whole-genome trio DNA sequencing revealed no candidate pathogenic genetic variants other than those detected using whole-exome sequencing, including miRNAs and lncRNAs.

#### 2.2.5. Imprinting Disorders and Multilocus Imprinting Disturbances Were Excluded by Methylation-Specific Multiplex Ligation-Dependent Probe Amplification

MS-MLPA targeting the 6q24.2, 7p12.2, 7q32.2, 11p15.5, 14q32.2, 15q11.2, 19q13.43, and 20q13.32 imprinted regions revealed no copy number or methylation abnormalities (Figure 7), thus excluding well-known imprinting disorder syndromes and multilocus imprinting disorders. We subsequently performed trio nanopore sequencing to assess the DNA methylation status of imprinted loci on chromosome 16.

#### 2.2.6. Imprinting Disorder on Chromosome 16 Revealed by Oxford Nanopore Sequencing

Oxford nanopore sequencing can detect DNA methylation from the ionic current signal of single molecules, offering a unique advantage over conventional methods [21]. We took advantage of this opportunity and performed nanopore sequencing of the proband’s and her parents’ DNA samples to characterize the methylation status of the known imprinted locus on chromosome 16. It was previously demonstrated that 16p13.3 encompasses the imprinted *ZNF597* gene [17]. In this case, we witnessed abnormal hypermethylation of the *ZNF597*:TSS-DMR, accompanied by hypomethylation of the neighboring *ZNF597*:3′ DMR (Figure 8).

## 3. Discussion

Trisomy 16 is one of the most common autosomal trisomies in humans [22]. Most cases of UPD (16) are consequences of trisomy rescue and are heterodisomic (UPhD). UPD (16) is known to be associated with trisomy 16 mosaicism, which might influence the phenotype of UPD (16) carriers. Another pathomechanism of UPD (16) is monosomy 16 rescue, where most cases are isodisomic (UPiD). Schematic representations of mechanisms and detectable genomic features for each UPD subtype can be found in previously published papers [23,24,25].

As a consequence of UPiD, homologous loci mapping to chromosome 16 are identical, and associated phenotypes may be due to unmasked mutations in recessive disease-related genes like *GPT2* [7], *FA2H* [8,9], *ABCA3* [10], *FANCA* [11], *SPG35* [12], *PMM2* [13], *ALG1* [14], *GAN* [15], and *WWOX* [16].

Due to the absence of causal pathogenic variants in the genes related to autosomal recessive diseases located on chromosome 16 and abnormal karyotype in our patient, we used case reports of patients with UPD (16) mat, excluding patients with either an effect of mutations of autosomal recessive genes or with an abnormal (including tissue mosaicism) postnatal karyotype, for a comparison of the UPD (16) mat and UPD (16) pat phenotypes. We also excluded UPD (16) mat patients with only prenatal ultrasound markers available without a postnatal/postmortem examination of the clinical presentation because, in these cases, information about the clinical outcome may be incomplete. The comparison of UPD (16) mat (excluding patients with either an effect of mutations of autosomal recessive genes or with only prenatal ultrasound markers available without a postnatal/postmortem examination of the clinical presentation, or with abnormal karyotype) [https://cs-tl.de/DB/CA/UPD/16-UPDm.html [26] (accessed on 30 June 2025); patients with UPD (16) pat; and our patient are presented in Table 1.

In 2019, Nakka et al. [23] provided an estimation of the prevalence of UPD in four million individuals from the general population and confirmed that UPD (16) is the most common UPD, with an overwhelming prevalence of UPD (16) mat. Among them, at least one-third are partial disomies not encompassing a known imprinted region at 16p13.3. In their study, the authors detected significant associations of UPD (16) pat with type 2 diabetes, hyperglycemia, and high cholesterol levels. It should be noted that the phenotypic features were ascertained from self-reported survey answers of 23andMe clients. The content of this survey, together with self-reporting, do not allow us to view the results as reliable and profound medical information. In particular, the survey lacks features descriptive for our patient with UPD (16) pat, such as intrauterine growth restriction, motor delay, microcephaly, high hairline, and arched eyebrows. Note that the authors clearly indicate the limitations of the 23andMe phenotypic data and of the conclusions drawn.

Patients with UPD (16) mat more often, but not always, have a Silver–Russel syndrome-like (SRS-like) phenotype [28,34,35,39], as opposed to UPD (16) pat, where patients have signs of general dysmorphogenesis, which is difficult to structure into a clear syndromic description due to the small number of observations and the prevalence of the normal phenotype [6,40]. In our patient, features such as preeclampsia and intrauterine growth restriction, combined with low physical development in the postnatal period, which are described as accompanying diseases of genomic imprinting at different stages of ontogenesis with the absence of causal variants in whole-exome sequencing data, allow us to assume that the patient’s phenotype is due to effects on the expression of imprinted genes located on chromosome 16 [41]. Since this patient with UPD (16) pat presents phenotypic manifestations different from UPD (16) mat, a different phenotypic effect of the expression of imprinted genes located on chromosome 16 can be assumed, depending on the parental origin. In our patient with UPD(16) pat, we observed DNA methylation abnormalities of the *ZNF597* gene DMRs that are consistent with the absence of a maternally derived copy of chromosome 16 in the proband. These changes are opposite to those previously described in UPD (16) mat [28], confirming that the imprinting anomalies in this region are due to UPDs of different parental origins. Moreover, at least one case has already been described of an isolated hypomethylation of the *ZNF597*:TSS-DMR and subsequent loss of imprinting in a patient with prenatal growth retardation and dysmorphic features, for whom UPD (16) was excluded [17], thus making it a candidate imprinting disorder syndrome. Evidence of imprinting at 16p13.3 recently provided a rationale for including analysis of copy number variations (CNVs) and methylation anomalies of *ZNF597* in the SALSA MLPA Probemix ME034-D1 Multi-Locus Imprinting Test [18]. The search for additional imprinted genes on chromosome 16 to explain the phenotype of alveolar capillary dysplasia with misalignment of pulmonary veins (ACDMPV) (OMIM# 265380), with deletions almost exclusively on the maternally inherited chromosome 16, showed evidence for novel candidate-imprinted loci on chromosome 16, namely, the maternally methylated DMR of *PRR25*, which is thought to be paternally expressed in supporting lymphoblastoid cells, and the paternally methylated DMR on 16q24.1 adjacent to LINC01082 mapping to the *FOXF1* enhancer [40].

Future observations of patients with UPD (16) pat will enable better differentiation of UPD (16) pat and UPD (16) mat phenotypes and highlight molecular genetic/epigenetic mechanisms.

## 4. Materials and Methods

### 4.1. Consent and Approval

All research participants gave their informed consent to clinical examination and the publication of their details and images (for the infant proband, the responsible adult signed a consent form). This study was performed in accordance with the Declaration of Helsinki and approved by the local ethics committee of the Research Center for Medical Genetics (approval number 2021-4/2).

### 4.2. Clinical Assessment

The family of the affected female were clinically examined at the Research Centre for Medical Genetics, Moscow, Russia.

### 4.3. Molecular Genetic Testing

Blood samples were collected from the proband and her unaffected parents, and genomic DNA was extracted using the phenol–chloroform procedure [42].

#### 4.3.1. Chromosomal Microarray Analysis

The CytoScan HD array (Affymetrix Inc., Santa Clara, CA, USA) was applied to detect CNVs across the entire genome according to the manufacturer’s protocols. Microarray-based copy number analysis was performed using Chromosome Analysis Suite software version 4.0 (Thermo Fisher Scientific Inc., Waltham, MA, USA), and the results were presented according to the International System for Human Cytogenomic Nomenclature 2016 (ISCN, 2016 [20]), https://karger.com/books/book/3554/ISCN-2016An-International-System-for-Human, accessed on 23 May 2025. All detected CNVs were assessed by comparing them with the published literature and the public databases, the Database of Genomic Variants (DGV), released 25 February 2020 (http://dgv.tcag.ca/dgv/app/home (accessed on 28 June 2022)), DECIPHER v11.31 (http://decipher.sanger.ac.uk/ (accessed on 28 June 2022)) and OMIM, updated 22 May 2025 (http://www.ncbi.nlm.nih.gov/omim (accessed on 28 June 2022)). Genomic positions refer to the human genome February 2009 assembly (GRCh37/hg19).

#### 4.3.2. Microsatellite Analysis

Seven microsatellite markers located along chromosome 16 (D16S3144, D16S403, D16S513, D16S2636, D16S3069, D16S3395, and D16S3399) were selected for microsatellite analysis to characterize the parental origin of the disomy and to investigate whether the UPD was UPiD or UPhD. Fragment analysis of the FAM-labeled PCR products was performed on an Applied Biosystems 3500 genetic analyzer (ThermoFisher Scientific, Waltham, MA, USA) according to the manufacturer’s instructions.

#### 4.3.3. Fluorescence In Situ Hybridization

FISH was performed on interphase nuclei and metaphase spreads from cells cultured from peripheral blood and skin fibroblasts, at least 300 each, according to the manufacturers’ protocols. DNA probes for the centromere region of chromosome 16 (SE 16 (D16Z2), Leica Biosystems, Amsterdam, The Netherlands) were applied. The analysis was performed using an AxioImager M1 epifluorescence microscope (Zeiss, Oberkochen, Germany) and an Isis digital image processing computer program (MetaSystems, Altlussheim, Germany) [43].

#### 4.3.4. Exome Sequencing

The patient’s DNA was analyzed using next-generation sequencing of 2 × 151 bp paired-end reads. The DNA library was enriched using a selective capture method targeting the protein-coding regions of human genes. The mean depth of coverage was 119× for this sample.

#### 4.3.5. Oxford Nanopore Sequencing

Whole genomic DNA was sequenced on an Oxford Nanopore PromethION instrument (Oxford Nanopore technology, Oxford, UK) equipped with R10.4.1 (Kit 14, Q20+) flow cells [44]. Signals in POD5 format were base-called with Dorado v0.9.1 using the *dna_r10.4.1_e8.2_400bps_hac@v5.0.0* model and *--min-qscore 15*; the integrated *5mCG_5hmCG v3* neural network simultaneously tagged 5-methyl- and 5-hydroxy-methyl-cytosines in CpG islands [45]. Reads were aligned to GRCh38 by the Dorado-embedded minimap2 engine using *-Y--secondary = yes* [46]. The resulting BAMs were coordinate-sorted with SAMtools v1.16.1 [47], and duplicate reads were flagged with MarkDuplicates from GATK v4.3.0.0 [48]. Germline SNVs and short indels were identified with Clair3 v1.0.11 (model r1041_e82_400bps_hac_v500) [49]. Haplotypes were reconstructed with WhatsHap v2.6, leveraging long-read phasing information [50]. Modified-base tags were converted from modBAM to bedMethyl and summarized with modkit v0.9.1 in pileup mode using the recommended parameters, *--filter-threshold C:0.75--filter-threshold A:0.85--interval-size 15,000,000--combine-strands--ignore h—cpg,* for downstream epigenetic analyses [51].

Further analysis and visualization were performed using R v.4.4.2 [52] with the vroom v.1.6.5 [53], dplyr v.1.0.0 [54], Gviz v.1.30.3 [55], and GenomicRanges v.1.38.0 packages [56]. The mean methylation levels of the parents’ DNA were used as the reference. The difference between the reference and proband was calculated using the formula*delta_i_* = *proband_i_* − *reference_i_*
where *delta_i_* is the difference between methylation levels in the reference and proband at the *i*-th locus. The *ZNF597* gene region is shown in Figure 8. We assumed deltas in the range [−0.2, 0.2] to be non-significant. These thresholds were chosen based on the average depth of sequencing of our samples and on other published works [57].

#### 4.3.6. Methylation-Specific Multiplex Ligation-Dependent Probe Amplification

MS-MLPA was performed using SALSA MLPA Probemix ME034-C1 Multi-locus Imprinting (lot: C1–0121, MRC-Holland, Amsterdam, The Netherlands), which analyzed the 6q24.2, 7p12.2, 7q32.2, 11p15.5, 14q32.2, 15q11.2, 19q13.43, and 20q13.32 imprinted regions. MS-MLPA was performed according to the manufacturer’s instructions. The results were analyzed using Coffalyser.Net™ software (v.250317.1029) developed and supported by MRC Holland.

## 5. Conclusions

We present a description of a 2-year-old female patient with a syndromic phenotype and UPiD (16) pat. This case report highlights the phenotype associated with the LOH of an insufficiently characterized imprinted region of the human genome. The accumulation of such observations will contribute to the systematization of data on the possible manifestations of imprinting disorders on chromosome 16. If methylation abnormalities are identified, the use of tools such as nanopore trio sequencing will help characterize understudied imprinted chromosomal regions such as 16p13.3.

## Figures and Tables

**Figure 1 ijms-26-08521-f001:**
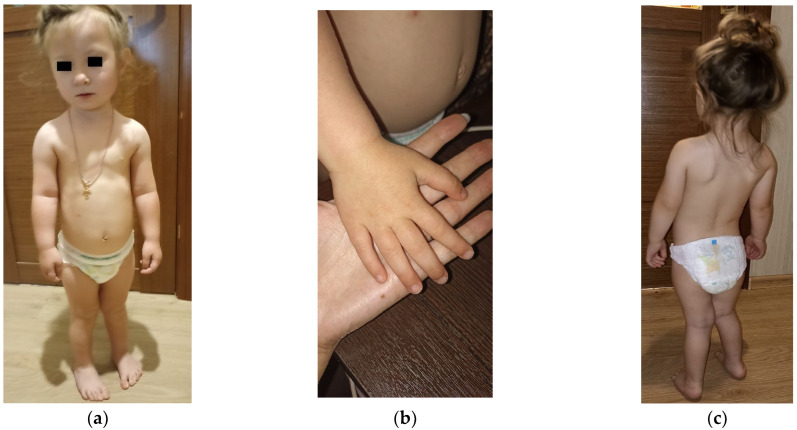
Patient’s phenotype included (**a**) microcephaly, high anterior hairline, arched eyebrows, hypotelorism, epicanthus, almond-shaped palpebral fissures, wide nasal bridge, depressed nasal ridge, wide base of the nose with a broad tip, smoothed filter, full cheeks, downturned corners of the mouth, short chin, dysplastic ears, narrow funnel-shaped chest, (**b**) cone-shaped fingers of the hands, proximal displacement of thumbs, (**c**) hyperlordosis, valgus knees and feet, and rocker-bottom foot.

**Figure 2 ijms-26-08521-f002:**
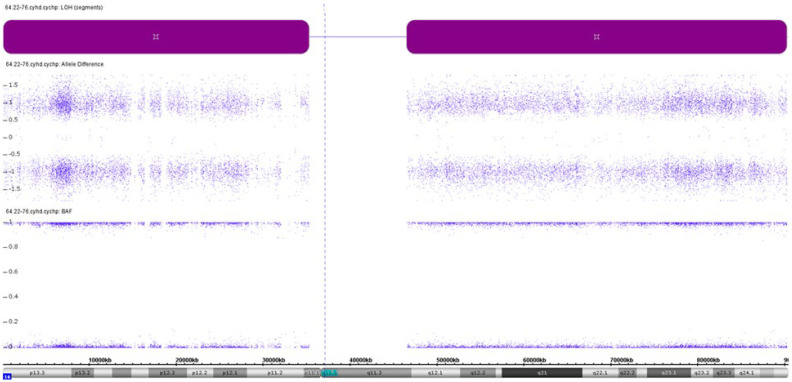
Chromosomal microarray analysis results showing a loss of heterozygosity (LOH) on chromosome 16 of the proband. LOH areas are presented as purple rectangles along the short and long arms of chromosome 16. The figure was generated using Chromosome Analysis Suite software version 4.0.

**Figure 3 ijms-26-08521-f003:**
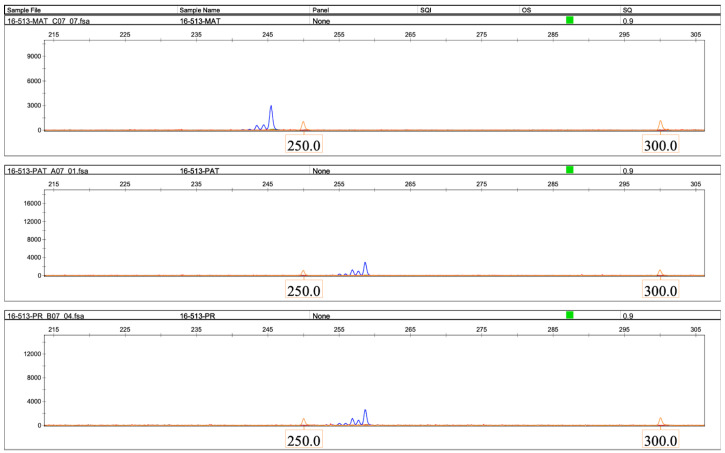
The result of microsatellite analysis using the D16S513 marker. (**Top panel**), maternal (MAT); (**middle panel**), paternal (PAT); (**bottom panel**), and proband (PR) DNA samples. X-axis, size of the microsatellite PCR product, bases. Y-axis, relative fluorescence units. The figure was generated using GeneMapper 6 software.

**Figure 4 ijms-26-08521-f004:**
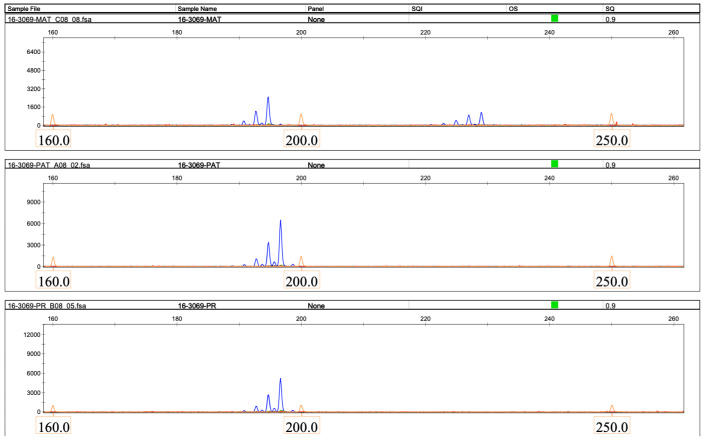
The result of microsatellite analysis using the D16S3069 marker. (**Top panel**), maternal (MAT); (**middle panel**), paternal (PAT); and (**bottom panel**), proband (PR) DNA samples. X-axis, size of the microsatellite PCR product, bases. Y-axis, relative fluorescence units. The figure was generated using GeneMapper 6 software.

**Figure 5 ijms-26-08521-f005:**
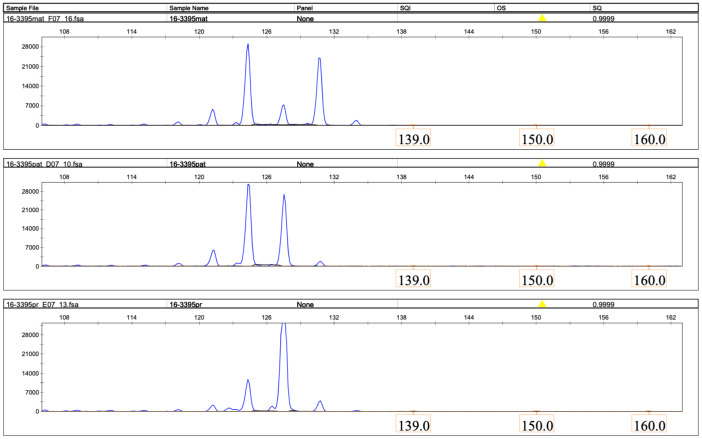
The result of microsatellite analysis using the D16S3395 marker. (**Top panel**), maternal (mat); (**middle panel**), paternal (pat); and (**bottom panel**), proband (pr) DNA samples. X-axis, size of the microsatellite PCR product, bases. Y-axis, relative fluorescence units. The figure was generated using GeneMapper 6 software.

**Figure 6 ijms-26-08521-f006:**
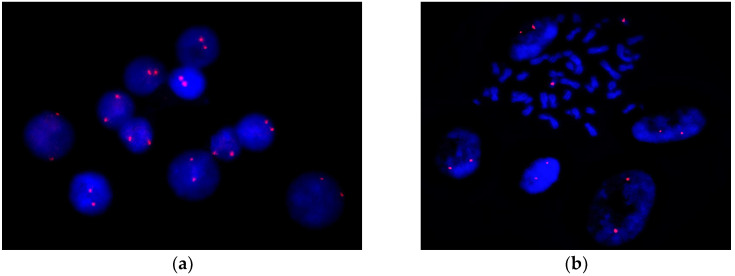
FISH results obtained with the D16Z1 DNA probe in cultured lymphocytes (**a**) and skin fibroblasts. (**b**) All cells demonstrate two hybridization signals corresponding to the two copies of chromosome 16. Image panels are at 1000× magnification.

**Figure 7 ijms-26-08521-f007:**
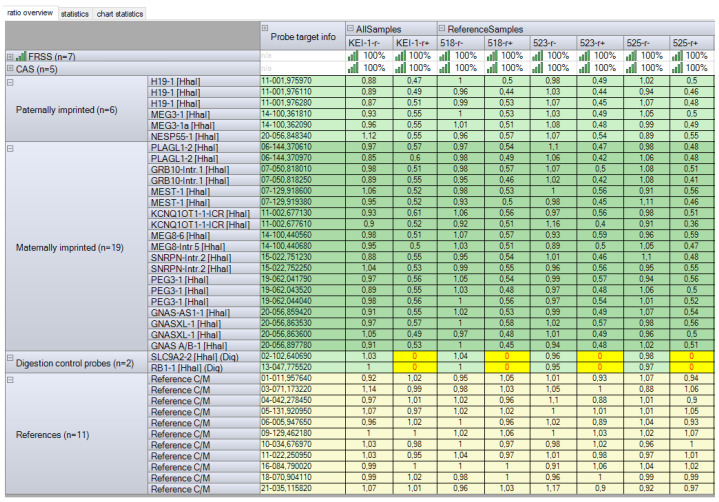
Results of the Methylation-Specific Multiplex Ligation-Dependent Probe Amplification analysis. “KEI-1”—test sample; “518”, “523”, and “525”—reference samples; “r−“—copy abnormality detected; “r+”—methylation abnormality detected. No copy abnormalities were detected, and the copy number status of the 6q24.2, 7p12.2, 7q32.2, 11p15.5, 14q32.2, 15q11.2, 19q13.43, and 20q13.32 imprinted regions was within 0.80–1.20. No methylation abnormalities were detected, and the methylation status of the 6q24.2, 7p12.2, 7q32.2, 11p15.5, 14q32.2, 15q11.2, 19q13.43, and 20q13.32 imprinted regions was within 0.40–0.65 (around 50% methylated, imprinted). The figure was generated using Coffalyser.Net™ software (v.250317.1029). FRSS, the Fragment Run Separation Score, is a quality measure of the analysis. CAS (Coffalyser Analysis Score) aggregates several different scores (including the FMRS—Fragment MLPA Reaction Score) to determine the quality of the comparative analysis, the percentage of quality of passing parameters in the reaction.

**Figure 8 ijms-26-08521-f008:**
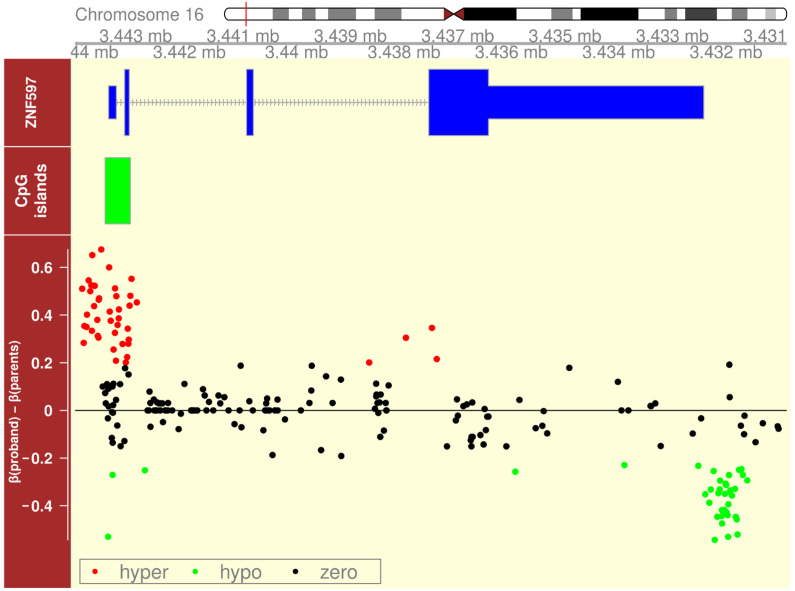
Imprinting disruption at two DMRs of the *ZNF597* gene in the proband. The Y-axis includes the upper track, showing a schematic of the *ZNF597* gene; the 5′ CpG island is colored green in the second track; and the lower track shows hypermethylation of the *ZNF597*:TSS-DMR (red points) and hypomethylation of the *ZNF597*:3′ DMR (green points), relative to the parents’ DNA methylation (black points). X-axis, the coordinates of the region under study on chromosome 16, bases. Y-axis, level of DNA methylation along the *ZNF597* gene. The result was obtained, and the figure was generated using R v.4.4.2.

**Table 1 ijms-26-08521-t001:** The comparison of UPD (16) mat, UPD (16) pat, and our patient’s phenotypes. N—number of patients. N/S—not studied in patients. Numbers in brackets refer to bibliographic references. The “+” sign indicates the presence of a symptom in the patient. The “−” sign indicates the absence of a symptom in the patient. * Fifteen patients (N = 1 [27] + N = 2 [28] + N = 1 [29] + N = 1 [30] + N = 1 [31] + N = 1 [32] + N = 1 [33] + N = 1 [34] + N = 2 [35] + N = 1 [36] + N = 1 [37] + N = 2 [38]). ** Two patients (N = 1 [5] + N = 1 [6]).

Features	Patients		
	UPD (16) Mat 15 Patients *	UPD (16) Pat 2 Patients **	UPD (16) Pat (Our Patient)
Prenatal			
short femora	1/15	−	**−**
abnormal echogenicity of the fetal left lower lung (possible isolated lung)	−	1/2	**−**
slight polyhydramnios	−	1/2	**−**
reverse flow in the ductus venosus	1/15	−	**−**
abnormal results of the maternal serum screen	3/15	1/2	**−**
abnormal results for the chromosome 16 number in maternal NIPT	N/S	1/2	**N/S**
chorionic villus sampling (CVS) karyotyping: trisomy 16 in all analyzed cells	7/15	−	**N/S**
amniotic fluid (AF) karyotyping: trisomy 16 mosaicism	3/15	1/2	**N/S**
placental mosaicism (CPM) for trisomy 16	−	1/2	**N/S**
amniotic fluid (AF) CMA: chromosome 16 LOH	N/S	1/2	**N/S**
maternal anemia	−	−	**+**
pre-eclampsia	2/15	−	**+**
maternal hematuria	1/15	−	**−**
maternal hepato-renal disfunction	1/15	−	**−**
two-vessel placenta	1/15	−	**−**
Infancy and childhood			
*general*			
PBL karyotyping: no trisomy 16 mosaicism or other chromosomal abnormalities	15/15	2/2	**+**
premature birth	8/15	1/2	**+**
intrauterine growth restriction	11/15	1/2	**+**
postnatal growth failure	9/15	1/2	**+**
feeding difficulties	6/15	−	**−**
muscular hypotonia	1/15	−	**−**
*abnormalities of the facial phenotype*			
protruding forehead	6/15	−	**−**
relative macrocephaly	4/15	−	**−**
microcephaly	1/15	−	**+**
triangular face	4/15	−	**−**
high anterior hairline	−	−	**+**
arched eyebrows	−	−	**+**
hypotelorism	−	−	**+**
epicanthus	1/15	−	**+**
almond-shaped palpebral fissures	1/15	−	**+**
slightly flatter face profile	1/15	−	**−**
wide nasal bridge	−	−	**+**
depressed nasal ridge	−	−	**+**
wide base of the nose with a broad tip	−	−	**+**
smoothed filter	−	−	**+**
full cheeks	−	−	**+**
downturned corners of the mouth	−	−	**+**
short chin	−	−	**+**
dysplastic ears	1/15	−	**+**
*abnormalities of the cardiovascular system*			
atrial septal defect	1/15	−	**+**
atrioventricular defect	4/15	−	**−**
ventricular septal defect (VSD)	2/15	−	**−**
hypertrophied, dilated right ventricle suggesting pulmonary hypertension	1/15	−	**−**
aortic stenosis	1/15	−	**−**
pulmonary valve stenosis	1/15	−	**−**
aortarctia	1/15	−	**−**
congenital heart disease	1/15	−	**−**
*abnormalities of the musculoskeletal system*			
scoliosis	2/15	−	**−**
body asymmetry	1/15	−	**−**
dislocation of the radio-humeral articulation	2/15	−	**−**
clinodactyly of the fifth fingers	7/15	−	**−**
assimilation of the atlas	−	−	**+**
spina bifida posterior C1	−	−	**+**
hypoplasia of the axial atlas	−	−	**+**
bilateral pes calcaneus	−	1/2	**−**
an additional rudimentary mandibular dental arch	−	1/2	**−**
narrow funnel-shaped chest	−	−	**+**
cone-shaped fingers of the hands	−	−	**+**
proximal displacement of thumbs	−	−	**+**
hyperlordosis	−	−	**+**
valgus installation of knees and feet	−	−	**+**
rocker-bottom foot	−	−	**+**
right talipes equinovarus	1/15	−	**−**
*abnormalities of the genitourinary system*			
hypospadias	3/15	−	**−**
pelvic dystopia	−	−	**+**
rotation of the left kidney	−	−	**+**
left hydronephrosis	1/15	−	**−**
left multi-cystic kidney	1/15	−	**−**
genitourinary anomalies	1/15	−	**−**
left renal agenesis	1/15	−	**−**
*abnormalities of the digestive system*			
esophageal atresia	1/15	−	**−**
tracheoesophageal fistula	1/15	−	**−**
giant cell hepatitis	1/15	−	**−**
inguinal hernia	1/15	−	**−**
*abnormalities of the respiratory system*			
pulmonary cystic changes	2/15	−	**−**
rudimentary bronchus on the right side	2/15	−	**−**
*abnormalities of the nervous system*			
delayed speech development	3/15	−	**+**
delayed motor development	1/15	−	**+**

## Data Availability

The datasets used and/or analyzed during this study are available from the corresponding author upon reasonable request.

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
