# Peer review of "Clinical and Molecular Presentation of a Patient with Paternal Uniparental Isodisomy of Chromosome 16"

_ijms, 2025, doi:10.3390/ijms26178521_

Round 1
Reviewer 1 Report
Comments and Suggestions for Authors
1. Introduction: I understand that there are rare cases, but there should be more references in the introduction. It is possible to specify the method by which these uniparental disomies are highlighted and when the first case was described regardless of the chromosome.
2. Results. 2.1. Clinical Presentation: This section is very detailed and provides enough information about the child's current situation, especially from a physical point of view. I think it could be completed with some concerns related to future development and a short management plan specifying the specialists who could be involved for a better recovery.
Figure 2. The type of microarray (SNP/CGH) must be specified for easier understanding
3. Discussion. It is stated that "Trisomy 16 is one of the most common autosomal trisomies in humans" but no reference is given here. I propose to add a paper that presents cytogenetic abnormalities in 330 Miscarriage Samples (DOI: 10.1159/000502304) with 26 WOS citations. The explanation of the mechanisms through which uniparental disomy can form are welcome and clearly presented. It would be even better if a scheme were added that would include all the variants for chromosome 16.
Table 1. The presentation of the case is on the last column. It should be marked in bold or gray to stand out.
4. Materials and Methods are explained in detail.
5. Conclusions are well outlined.
Author Response
Dear Reviewer, the response file is attached.

Reviewer 2 Report
Comments and Suggestions for Authors
Authors report another case of an isoUPD(16)pat.
Acc. to https://cs-tl.de/DB/CA/UPD/0-Start.html there are at least 118 cases with UPD(16)mat, 15 cases with UPD(16)pat and 45 cases in which parental origin of UPD(16) was not determined.
As shown by Nakka et al. 2019 (PMID: 31607426) UPD(16) is the most frequent UPD in normal human population.
It is clear since that that UPD(16) does not cause any harm as it is no imprinted chromosome. This reference is not cited in the paper.
Accordingly, the presented case must have some other problem - most likely not related to UPD(16)pat.
Before paper may be possibly be accepted authors need to check if the patient may not suffer from a MLID or another imprinting disease related disorder - performed studies do not exclude a heterodisomy of chromosomes 7 or 15. This needs to be tested as face2gene gave hints on these disorders.
Overall, the case must be considered as an unsolved case with UPD(16)pat, which definitely is not disease causing. So this is not enough to diserve a publiccation.
Author Response

(The authors gave the same response as above.)

Reviewer 3 Report
Comments and Suggestions for Authors
The manuscript deals with the clinical and molecular characterization of a syndromic child found to have paternal uniparental disomy 16 (UiPD). The report is interesting due to the clinical and molecular characterization in the literature of only 2 cases of pat UPD 16, both of the isodisomy type. The Authors describe in detail the clinical phenotype of the girl in comparison to the less numerous and severe clinical features reported in the two published cases. By chromosomal microarray they reveal chromosome 16 UPD and by chromosome 16 specific microsatellite analysis show that the girl harbors the pat allele at three chromosome 16 microsatellites (Fig. 2). By WES results the impact on the phenotype of chromosome 16 pathogenic variants causing recessive diseases via 16 UiPD is ruled out. I suggest to mention also WES reads at chromosome 16 polymorphic loci to support the key result of pat disomy 16 . Major criticism: lack of the differential methylation of CpGs at known chromosome 16 paternal imprinted genes (https://www.geneimprint.com/site/genes-by-species) . WES analysis should be expanded by DNA methylation analysis to search for a methylation defect . As shown by Gonench Kilich et al, 2024 ( https://doi.org/10.1038/s41525-023-00389-2) this analysis does not need any more high-resolution bisulfite sequencing (Schulze, 2019; Yamazawa, 2021) but only a commercial kit, making it easy to use and worthwhile to unravel the severe and complex phenotype of the described unique case. Whatever the result, this further assessment will reinforce the conclusions of the study.
Minor criticisms
Abstract line 19 I would replace “several” with a few
I suggest to include WES in the key words
Line 111 trisomy rescue may replace trisomic rescue
Table 1 is quite informative. However the number of UPD 16 is likely underestimated.
Author Response

(The authors gave the same response as above.)

Reviewer 4 Report
Comments and Suggestions for Authors
In manuscript ijms-3304477 the authors describe a clinical case of a young girl showing a complex phenotype and associate this to a paternal uniparental isodisomy of chromosome 16. Overall, the research is of fair quality, but it needs some improvements before publication.
Introduction. This section is very short and does not present all the data needed to fully understand the obtained results. In particular, since most of the conclusions involve chromosome imprinting, I strongly suggest to expand this part, by generally describing human imprinting (see for example Carli et al 2020, doi: 10.4274/jcrpe.galenos.2019.2018.0249) and specifically chr16 imprinting (see for example the Carli’s review, but also this manuscript: Yong et al, 2002, https://pubmed.ncbi.nlm.nih.gov/12244544/).
Results. In general, every subsection is composed of 1-2 sentences; this shortness does not justify their separation into independent paragraphs, also because they basically describe different aspects of the same thing (i.e., chr16 isodisomy). Consider merging them into one section. In addition, the comparison of UPD(16)mat, UPD(16)pat and proband’s phenotypes presented in Table 1 sounds to me more a result than a discussion; consider moving this part into the Results, creating an independent paragraph dealing with clinical data.
Table 1. Data in column 2 are hard to read, it would be interesting to see if these many defects are not uniformly distributed among patients, i.e., they accumulate in a minority of the reported probands. I suggest expanding the table, and create a column for each proband.
Discussion. In lines 171-172 the authors interpret their results by writing that data “allow us to assume that the result of the patient's phenotype is due to affected expression of imprinted genes located on chromosome 16 [13].” However, this is not the only possible explanation. The authors analyzed the exome of the proband, yet this does not exclude the presence of unidentified mutations in genes other than protein-coding ones, such as miR or lncRNA. See for example Boroumand et al 2019, DOI : 10.1186/s43042-019-0041-2 (for miR) and https://www.lncipedia.org/ (for lncRNA mapping on chr16). I suggest to add this part as an alternative explanation of the phenotype of the proband – unless the authors are willing to analyze also the transcriptome of chr16 related to non-coding RNAs.
Materials and Methods: in lines 198-199, accession dates and/or database version should be added for all data interrogations.
Conclusions. See also comments on Discussion. In addition, the previously suggested work of Carli et al 2020 reports that "as for UPD(6)mat, (...) a specific chromosome 16 associated imprinting disorder does not exist (105)". Authors should comment on this.
Minor points.
· Line 41: 1-year-old – did you mean 2-years-old?
· Line 49: weight “6.5” refers to 11 months of age? It seems referred to 8 months. Please check the language.
· Line 62: Face2Gene service; please add accession date and software version. Also, add this service in the bibliographic references.
· Lines 75 and 196: please add bibliographic reference, available at https://karger.com/books/book/3554/ISCN-2016An-International-System-for-Human.
· Lines 122 and following (Table 1): I suppose that numbers in brackets refer to bibliographic references; if so, please disclose this in Table caption. In addition, also disclose the meaning of plus and minus signs.
Author Response

(The authors gave the same response as above.)

Round 2
Reviewer 2 Report
Comments and Suggestions for Authors
Authors ignored some parts of previous comments
so here again:
Authors report another case of an isoUPD(16)pat.
They cite now the database https://cs-tl.de/DB/CA/UPD/0-Start.html
But they need to cite as given on the webpage as Liehr T. 2025. Cases with uniparental disomy. https://cs-tl.de/DB/CA/UPD/0-Start.html [accessed on ...]
They select in a not clear way several cases from there but neither mention that there are at least 118 cases with UPD(16)mat, 15 cases with UPD(16)pat and 45 cases in which parental origin of UPD(16) was not determined.
Also the Nakka et al. 2019 (PMID: 31607426) but do not mention that UPD(16) is the most frequent UPD in normal human population.
It is clear since that that UPD(16) does not cause any harm as it is no imprinted chromosome. Nonetheless authors insist with giving at in introduction 3 references that there shall be a known imprinted region at 16p13.3. After checking this literature there is not clear evidence for this statement - it is all suggestions in these papers. All symptoms of the patients can be explained by trisomy in placenta. There is no evidence that there exists an imprinting disorder in connection with UPD(16).
Accordingly, the presented case must have some other problem - most likely not related to UPD(16)pat.
Overall, the case must be considered as an unsolved case with UPD(16)pat, which definitely is not disease causing.
Besides, authors refer to ISCN 2016; but it must be referred now to ISCN 2024
Author Response
Dear Reviewer, thank you very much for your help in improving our manuscript.
Reviewer 3 Report
Comments and Suggestions for Authors
Demonstration- by nanopore sequencing- in the UPiD(16) proband of imprinting disruption of the ZNF597 gene DMRs opposed to those described for UPD(16) mat is the valuable conclusion of the reported case. It will be the steady reference for future cases! The manuscript has also greatly improved in its general structure.
Minor discretionary points : the readability of fig.7 is scarce: this Fig may become Suppl
Pay attention to the text erroneously aligned to Table 1 (page 19) and add the reference number after Nakka et al
Author Response

(The authors gave the same response as above.)

Reviewer 4 Report
Comments and Suggestions for Authors
Most of my comments were adequately addressed by authors, the remaining being mostly aestethic and not substantial. As long as they are happy with that form, I'm fine too.
Author Response

(The authors gave the same response as above.)
